# Evaluation of the Antibacterial Activity and Efflux Pump Reversal of Thymol and Carvacrol against *Staphylococcus aureus* and Their Toxicity in *Drosophila melanogaster*

**DOI:** 10.3390/molecules25092103

**Published:** 2020-04-30

**Authors:** Zildene de Sousa Silveira, Nair Silva Macêdo, Joycy Francely Sampaio dos Santos, Thiago Sampaio de Freitas, Cristina Rodrigues dos Santos Barbosa, Dárcio Luiz de Sousa Júnior, Débora Feitosa Muniz, Lígia Claudia Castro de Oliveira, José Pinto Siqueira Júnior, Francisco Assis Bezerra da Cunha, Henrique Douglas Melo Coutinho, Valdir Queiroz Balbino, Natália Martins

**Affiliations:** 1Laboratory of Semi-Arid Bioprospecting (LABSEMA), Regional University of Cariri-URCA, Crato 63105-000, CE, Brazil; Zildenesousa15@gmail.com (Z.d.S.S.); naiirmacedo@gmail.com (N.S.M.); joycy.sampaio22@gmail.com (J.F.S.d.S.); darciolsjr@gmail.com (D.L.d.S.J.); ligiaclaudia@yahoo.com.br (L.C.C.d.O.); cunha.urca@gmail.com (F.A.B.d.C.); 2Graduate Program in Biological Sciences-PPGCB, Federal University of Pernambuco-UFPE, Recife 50670-901, PE, Brazil; vqbalbino@gmail.com; 3Laboratory of Microbiology and Molecular Biology (LMBM), Regional University of Cariri-URCA, Crato 63105-000, CE, Brazil; thiagocrato@hotmail.com (T.S.d.F.); cristinase75@gmail.com (C.R.d.S.B.); deehmuniz78@gmail.com (D.F.M.); 4Laboratory of Microorganism Genetics (LGM), Federal University of Paraiba-UFPB, João Pessoa 58051-900, PB, Brazil; siqueira@dbm.ufpb.br; 5Faculty of Medicine, University of Porto, Alameda Prof. Hernâni Monteiro, 4200-319 Porto, Portugal; 6Institute for Research and Innovation in Health (i3S), University of Porto, 4200-135 Porto, Portugal

**Keywords:** bacterial resistance, efflux pumps, terpenoids, thymol, carvacrol

## Abstract

The antibacterial activity and efflux pump reversal of thymol and carvacrol were investigated against the *Staphylococcus aureus* IS-58 strain in this study, as well as their toxicity against *Drosophila melanogaster*. The minimum inhibitory concentration (MIC) was determined using the broth microdilution method, while efflux pump inhibition was assessed by reduction of the antibiotic and ethidium bromide (EtBr) MICs. *D. melanogaster* toxicity was tested using the fumigation method. Both thymol and carvacrol presented antibacterial activities with MICs of 72 and 256 µg/mL, respectively. The association between thymol and tetracycline demonstrated synergism, while the association between carvacrol and tetracycline presented antagonism. The compound and EtBr combinations did not differ from controls. Thymol and carvacrol toxicity against *D. melanogaster* were evidenced with EC_50_ values of 17.96 and 16.97 µg/mL, respectively, with 48 h of exposure. In conclusion, the compounds presented promising antibacterial activity against the tested strain, although no efficacy was observed in terms of efflux pump inhibition.

## 1. Introduction

The prevalence of bacterial resistance to antibiotics, this being associated with increased mortality rates, has become a source of great concern for public health [1]. *Staphylococcus aureus* is a commensal microorganism associated with a wide variety of infections, since it has the capacity to acquire resistance to many classes of antibacterial agents, such as β-lactams, quinolones and macrolides [2,3].

There are several mechanisms by which *S. aureus* develops resistance to antimicrobials, including limited drug absorption, target modification, enzymatic inactivation and active efflux mechanisms [4]. The latter, also known as efflux pumps, are proteins integrated into the bacterial plasma membrane that reduce the intracellular concentration of antibiotics by extruding them from the cell [5]. Among these pumps, the TetK pump, belonging to the major facilitator superfamily (MFS), is present in *S. aureus* IS-58 strain. TetK powers its transport activity with energy derived from proton gradients and is responsible for resistance to the tetracycline class of antibiotics [6].

Given the above, the development of efflux pump inhibitors that act as competitive and non-competitive adjuvants to reduce antibiotic resistance has attracted the attention of researchers [7]. Natural bacterial resistance modifiers can facilitate the reintroduction of ineffective therapeutic antibiotics in the clinic, reducing the toxic risks of these drugs by acting as efflux system regulators or as efflux pump inhibitors (EPIs) when blocking their activity [8,9].

The compounds thymol and carvacrol are two phenolic terpenoids, geometric isomers, which can be found in the form of translucent crystals and a yellowish liquid, respectively, at room temperature. Both are obtained from essential oils, mainly from thyme (*Thymus vulgaris* L.) and oregano (*Origanum vulgare* L.) [10], where a number of pharmacological properties associated with these compounds have been previously described in the literature, including antifungal [11] and antibacterial activities [12,13]. Although some compounds are capable of acting as EPIs, their high eukaryotic cell toxicity prevents their development as EPIs [14]. Thus, studies assessing the toxicity of these substances are necessary. *Drosophila melanogaster* is a model organism in toxicological assays which aim to understand the genetic and molecular mechanisms of toxic substances since these are very sensitive to different concentrations of toxic substances [15].

Thus, the objective of this study was to assess the antibacterial activity and efflux pump reversal mechanisms of the isomers thymol and carvacrol against the *S. aureus* IS-58 bacterial strain and to evaluate their toxicity in a *D. melanogaster* model.

## 2. Results

### 2.1. Minimum Inhibitory Concentration (MIC)

The monoterpenes thymol and carvacrol demonstrated relevant direct antibacterial activity against the *S. aureus* IS-58 strain, with MIC values of 72 µg/mL and 256 µg/mL, respectively (Table 1), where the MIC value for thymol was more effective than that of the standard antibiotic tetracycline, with values varying between 128 and 114 µg/mL.

### 2.2. Modulatory Effect over Antibiotic Activity and Ethidium Bromide

When thymol was combined at a sub-inhibitory concentration (MIC/8) with tetracycline, an interference of the antibiotic activity was observed, where a MIC reduction from 114 to 101 µg/mL was seen (Figure 1A). When the antibiotic was tested in association with standard inhibitors, the MIC values for chlorpromazine did not differ from the control, whereas in association with CCCP, a reduction in the antibiotic MIC was observed, indicating greater specificity of this inhibitor for the tested pump, with inhibition of the antibiotic resistance mechanism being observed.

With respect to efflux pump inhibition, the assays with ethidium bromide (EtBr) as a pump substrate found the association between thymol and EtBr did not differ from the control. Thus, the observed action of the compound, when in association with the antibiotic, suggests an activity over a resistance mechanism other than active efflux (Figure 1B).

The data regarding the combined effect of carvacrol and tetracycline reported an antagonism, with the MIC value increasing from 128 to 203 µg/mL (Figure 2A). However, the results for the association of carvacrol with chlorpromazine did not differ from the control, while a synergism resulting from its association with carvacrol was observed for CCCP.

When the EPI effect was evaluated using the MIC reduction of EtBr by carvacrol, the association was shown to not differ from the control, while standard inhibitors synergistically modulated this, demonstrating the presence of an efflux pump mechanism (Figure 2B).

The results observed in the graphs above indicate the presence of an efflux mechanism that is sensitive to the CCCP inhibitor when tested in association with the antibiotic against the *S. aureus* IS-58 strain. Furthermore, the association with EtBr demonstrated presence of an efflux pump mechanism sensitive to both standard inhibitors, which shows that the antibacterial activity exhibited by both thymol and carvacrol against *S. aureus* is not due to an EPI effect.

### 2.3. Drosophila Melanogaster Toxicity Assay and Negative Geotaxis

The monoterpenes thymol and carvacrol presented marked toxicity against *D. melanogaster* with EC_50_ values of 17.96 µg/mL and 16.97 µg/mL, respectively, within 48 h of exposure to the products. The mortality tests using thymol found the highest concentration tested, 31 µg/mL, and caused significant mortality compared to the control and the other concentrations from the first reading performed, which corresponded to 3 h of exposure to the compound (Figure 3A).

In the 6 to 48-h readings, the concentration-dependent toxicity pattern remained constant, where the 8 and 15 µg/mL concentrations did not show significant differences compared to the control with respect to the number of dead flies. For carvacrol, a toxicity for this compound was found at 30 μg/mL, this being the most potent, since it began to differ statistically from the control after the first 6 h of exposure, becoming more effective throughout the exposure readings, showing a high mortality rate in the flies. The 7 and 15 μg/mL concentrations were not significant compared to the control (Figure 3B). Damage to locomotor capacity following exposure to thymol was determined by the negative geotaxis test, in which the 31 µg/mL concentration was found to cause damage to the *D. melanogaster* locomotor apparatus following 3 h of exposure to the compound, this being statistically significant compared to the control and the other concentrations (Figure 4A). The mobility of the flies was scarcely affected by the 8 and 15 µg/mL concentrations, with no statistically significant interferences in fly locomotion compared to the control group.

In the locomotor system damage assessment, a marked decline in the behavioral response of the flies was observed when these were exposed to a 30 μg/mL concentration, from the first 3 h of exposure to the compound. This effect was intensified in the following hours, such that by the 24-h reading the live flies showed great locomotor difficulties (Figure 4B).

## 3. Discussion

In this work, our results indicated that thymol and carvacrol exert relevant antibacterial activity, with MIC values of 72 and 256 µg/mL respectively, against the *S. aureus* IS-58 strain. These results are in accordance with that reported by Miladi et al. [16], in which thymol and carvacrol obtained MIC values of 64 and 256 µg/mL, respectively, against the *S. aureus* ATCC 25923 strain. Lambert et al. [17] also reported a greater antibacterial activity for thymol compared to carvacrol against *S. aureus* ATCC 6538. These results may be justified by the hydrophobic nature and low solubility of thymol in the hydrophobic domain of the cytoplasmic membrane of bacterial cells. [18].

The structural differences in the hydroxyl group (OH) position in both thymol and carvacrol isomers did not affect the inhibitory effect against the assayed bacterial strain. By the same reason, the relative position of the hydroxyl group in the phenolic ring also failed to strongly influence the degree of antibacterial activity for thymol and carvacrol against *Bacillus cereus*, *S. aureus* and *Pseudomonas aeruginosa* [19]. However, the study by Dorman and Deans [20] reported that carvacrol and thymol act differently against gram-positive and gram-negative species. Our results evidenced an interference in the antibiotic activity when thymol and tetracycline were associated, resulting in a reduction of the MIC value of tetracycline from 114 to 101 µg/mL. However, Davies and Wright [21] stated in their study that when a compound is in association with an inhibitor, only a minimum of a 3-fold MIC reduction is acceptable as significant in terms of inhibiting resistance mechanisms, when carvacrol was associated with tetracycline an antagonism can be observed. These results are different from that observed by Cirino et al. [22], where thymol and carvacrol, used at sub-inhibitory concentrations (MIC/4), reduced the MIC value of tetracycline from 64 μg/mL to 32 μg/mL in both cases.

The TetK efflux pump is the main mechanism for bacterial resistance to tetracycline, being coded by the plasmid gene pt181. The main efflux protein protects the bacterial ribosome by extruding the antibiotic out of the bacterial cell [23]. Many studies have been conducted to face the bacterial resistance to antibiotics, mainly through using compounds that act as adjuvants to the antibiotic activity. For this reason, it is known that natural products and phytochemicals act synergistically with this objective [24,25,26]. In our assays, thymol and carvacrol had no effect as EPIs. Due to the fact that efflux pumps are the unique mechanism for EtBr extrusion, the MIC reduction of the EtBr indicates an EPI effect [27]. Thus, our results indicate that thymol and carvacrol act on other resistance mechanisms, regardless of the active efflux.

Given these results, we also investigated the thymol and carvacrol toxicity against *D. melanogaster*. Similar findings were also stated by Zhang et al. [28], who demonstrated that oxygenated monoterpenes, such as thymol and carvacrol, exhibited high toxicity against *D. melanogaster* while investigating the fumigant toxicity of monoterpenes against fruit flies. Negative geotaxis consists of flies’ ability to move vertically, this being a common locomotion behavior associated with *D. melanogaster* [29]. The results found in this study corroborate data from the investigation by Karpouhtsis et al. [30], which reported an insecticidal and genotoxic activity for thymol against *D. melanogaster*. Previous studies also report a repellent activity for thymol against *Culex pipiens pallens* [31], as well as a toxic effect for its larvae [32].

The insecticidal potential of terpenoids has been associated with their low molecular weights, which makes them highly volatile, with these being often considered toxic or repellent against insects, where different functional groups and their locations seem to influence their biological effectiveness [33]. The toxicity mechanisms and locomotor capacity impairments produced by many compounds and/or essential oil may be associated with a decrease in AChE activity, which, considering its importance against neurotoxicity, functions as defense in stressful situations [34,35]. Another factor also reported in the literature is the increase in the biosynthesis of heat shock proteins, such as *hsp70*, which support the functional structure of important enzymes and proteins, as an insect’s self-defense mechanism against stressors [36].

## 4. Materials and Methods

### 4.1. Bacterial Strain and Culture Media

The IS-58 *S. aureus* strain (gently furnished by Dr. Simon Gibbons, from the Imperial College, London, UK), with the PT181 plasmid carrying the gene for the tetracycline efflux protein TetK, was used. The culture media used in the tests were heart infusion agar (HIA, laboratories Difco Ltd.a., Campinas, Brazil) prepared according to the manufacturer and 10% brain heart infusion (BHI, laboratories Difco Ltd.a.) broth.

### 4.2. Substances

The antibiotic tetracycline, as well as thymol and carvacrol, were diluted in dimethyl sulfoxide (DMSO) and in sterile water to a final concentration of 1024 μg/mL. The DMSO proportion used was lower than 5%. Chlorpromazine and EtBr were dissolved in sterile distilled water, while carbonyl cyanide m-chlorophenylhydrazone (CCCP) was dissolved in methanol/water (1:3, *v*/*v*). All substances were diluted until reaching a concentration of 1024 µg/mL. The molecular structures of the compounds were obtained using the software ACD/ChemSketch (ACD/LABS, Toronto, ON, Canada) (Figure 5).

### 4.3. Determining the Minimum Inhibitory Concentration (MIC)

The MIC was determined for the isolated compounds thymol and carvacrol as per the broth microdilution method proposed by Javadpour et al. [37], with some adaptations. The inoculants were prepared 24 h after sowing the strains. Eppendorfs^®^ were filled with 1440 µL of BHI and 160 µL of the inoculum. The plates were then filled with 100 µL of the final solution. Microdilution was performed with 100 µL of the products. Following 24 h of incubation, readings were taken by the addition of resazurin (7-hydroxy-3H-phenoxazine-3-one 10-oxide) [38]. The tests were performed in triplicates.

### 4.4. Evaluation of Efflux Pump Inhibition

Efflux pump inhibition was carried out using the methodology adapted from Coutinho et al. [39]. Eppendorfs^®^ were filled with 160 µL of the inoculum, the sub-inhibitory concentration (MIC/8) of the compounds, and completed with BHI until reaching a volume of 1.6 mL. Microdilution was performed with 100 µL of the antibiotics, and readings were taken 24 h after incubation by adding resazurin [38]. The modulatory effect of the combination of the antibiotic, as well as EtBr, with the compounds’ thymol and carvacrol was tested using a methodology adapted from Coutinho et al. [39] (Figure 5). For this, Eppendorfs^®^ were filled with 160 µL of the inoculum with the compounds at sub-inhibitory concentrations (MIC/8) and completed with BHI until reaching a volume of 1.6 mL. A modulatory control was prepared with 160 µL of the inoculum and 1440 µL of BHI. Thereafter, the microdilution plates were filled, with rows G and H corresponding to the microbial growth controls. Sterility controls were performed on separate plates. Subsequently, microdilutions were performed with the antibiotic and EtBr (100 µL). After 24 h, readings were taken in the same manner as for the MIC tests. The tests were performed in triplicates.

### 4.5. Drosophila melanogaster Toxicity Assays

The fumigation bioassay method proposed by Cunha et al. [40] was used to assess toxicity. Adult flies (males and females), in multiples of 20, were placed in 130 mL flasks, previously prepared with 1 mL of 20% sucrose solution. The compound doses were impregnated in the glass cover on filter paper. The control received 20 µL of acetone, while the compounds were prepared at the concentrations of 200 µg/mL for thymol and 195.2 µg/mL for carvacrol, both diluted in acetone. Volumes of 20, 10 and 5 µL were taken from the stock solutions, resulting in the final concentrations of 31, 15 and 8 µg/mL in bottles with 130 mL of air for thymol and 30, 15 and 7 µg/mL in bottles with 130 mL of air for carvacrol, respectively.

The bioassays were conducted in a BOD-type greenhouse under controlled conditions. The tests were performed in triplicates and mortality rate readings were performed at 3, 6, 12, 24, 36 and 48 h [40].

### 4.6. Negative Geotaxis Assays

Damage to the locomotor system was determined as described by Coulom and Birman [41], after considering fly mortality. The negative geotaxis assay consists of counting the number of flies that rise above 3 cm in the experimental glass column during a 5 s time interval. The assays were repeated twice within a 1 min interval. The results were presented as the mean time (s) ± SE obtained from three independent experiments.

### 4.7. Statistical Analysis

A two-way analysis of variance (ANOVA) followed by Bonferroni’s post hoc test was employed for the microbiological assays using GraphPad Prism 7.0 software (GraphPad Software, San Diego, CA, USA). Meanwhile, a two-way ANOVA followed by Tukey’s multiple comparisons test was performed for the toxicity data analysis.

## 5. Conclusions

The monoterpenes thymol and carvacrol presented direct antibacterial activity against the *S. aureus* IS-58 strain, where the strain was shown to be more sensitive to thymol, with the isomeric difference being a possible factor in terms of antibacterial activity. However, despite the demonstrated antibiotic activity results, the compounds were ineffective at inhibiting the TetK efflux pump mechanism, thus indicating that the antibacterial activity of the compounds is not associated with this resistance mechanism. Thymol and carvacrol exerted a marked toxic activity against *D. melanogaster*.

## Figures and Tables

**Figure 1 molecules-25-02103-f001:**
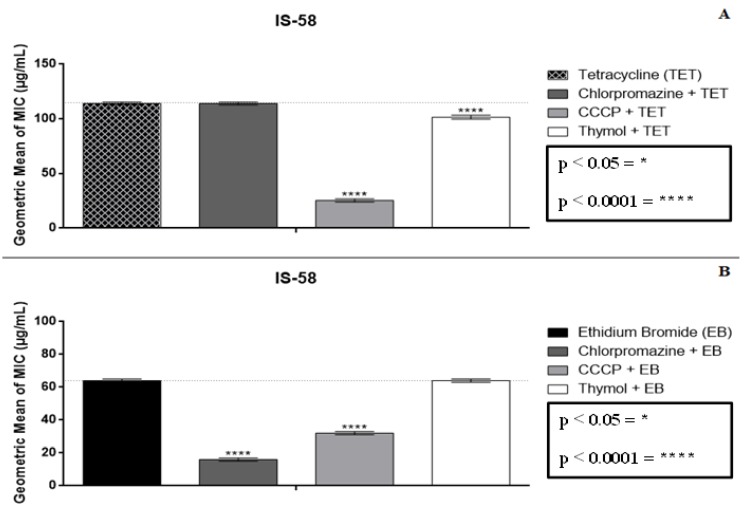
Effect of the association between thymol and tetracycline (**A**) and thymol and ethidium bromide (**B**) over *S. aureus* IS-58, expressing the TetK efflux protein. CCCP = carbonyl cyanide m-chlorophenylhydrazone; * *p* <0.05; **** *p* < 0.0001.

**Figure 2 molecules-25-02103-f002:**
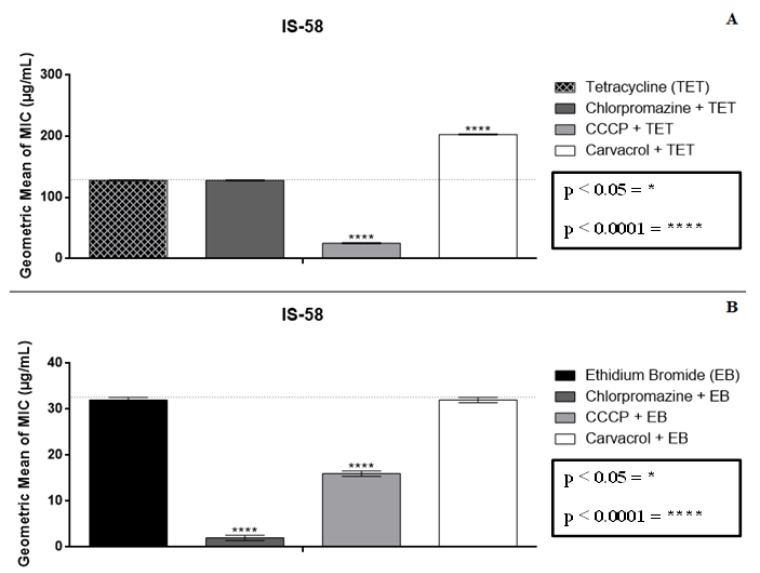
Effect of the association between carvacrol and tetracycline (**A**) and carvacrol and ethidium bromide (**B**) over *S. aureus* IS-58, expressing the TetK efflux protein. CCCP = carbonyl cyanide m-chlorophenylhydrazone; * *p* < 0.05; **** *p* < 0.0001.

**Figure 3 molecules-25-02103-f003:**
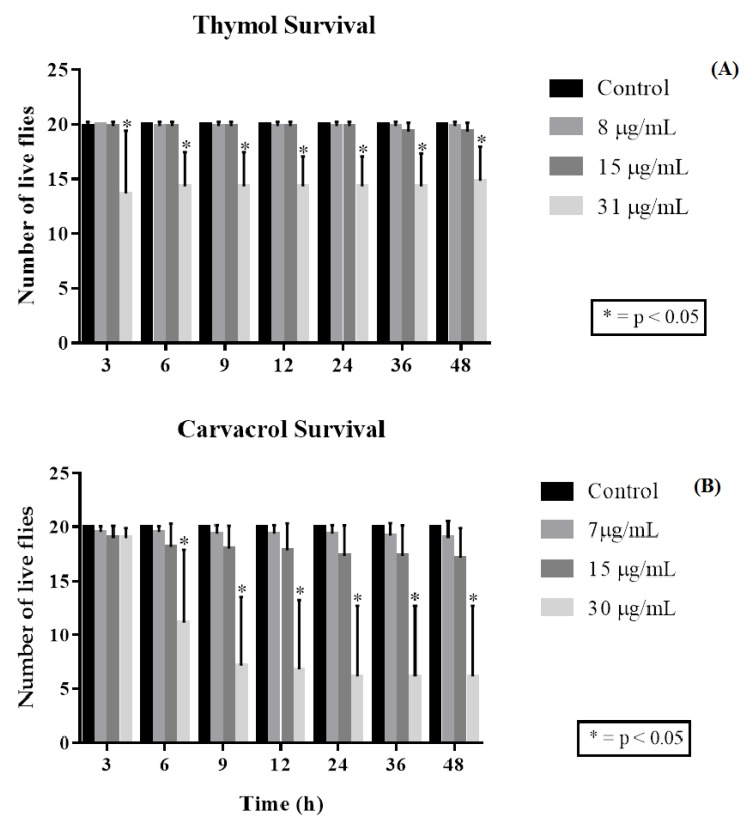
Toxic effect of different thymol (**A**) and carvacrol (**B**) concentrations on *D. melanogaster*.

**Figure 4 molecules-25-02103-f004:**
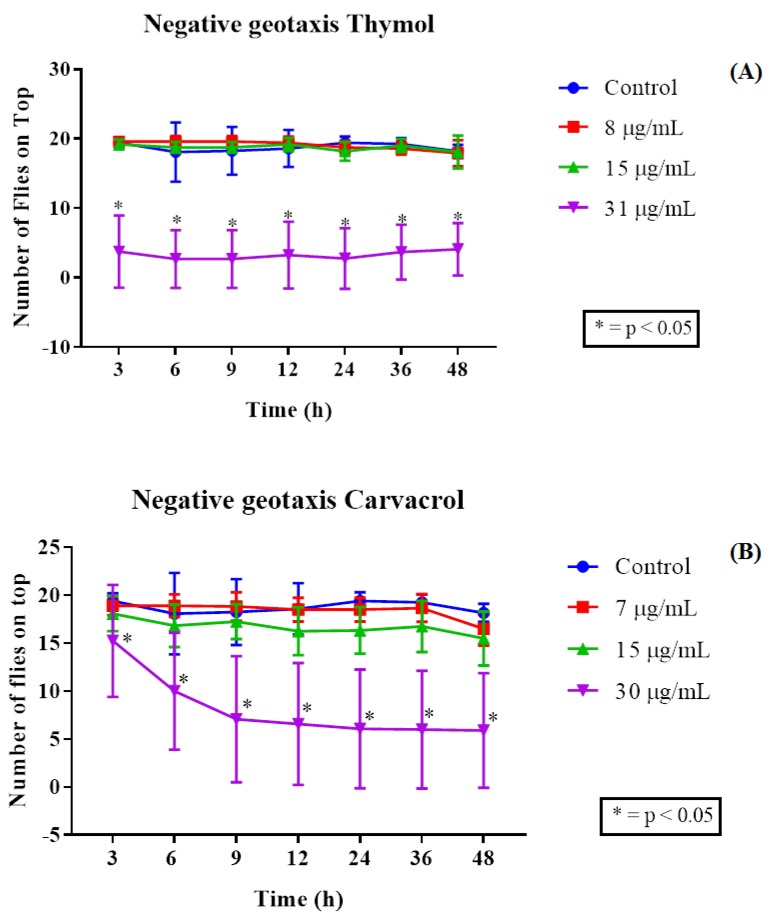
Toxic effect of varying thymol (**A**) and carvacrol (**B**) concentrations on the locomotor ability of *D. melanogaster*.

**Figure 5 molecules-25-02103-f005:**
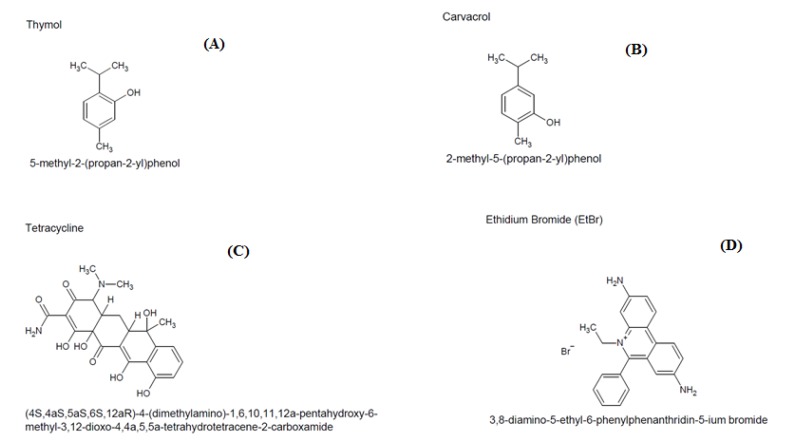
Chemical structures of thymol (**A**), carvacrol (**B**), tetracycline (**C**) and ethidium bromide (**D**).

**Table 1 molecules-25-02103-t001:** Minimum inhibitory concentrations (MIC, μg/mL) of thymol, carvacrol and tetracycline against the *S. aureus* IS-58 strain.

Strain	MIC (µg/mL)		
*S. aureus* IS-58	Thymol	Carvacrol	Tetracycline
	72	256	128

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
