# Peer review of "Evaluation of the Antibacterial Activity and Efflux Pump Reversal of Thymol and Carvacrol against Staphylococcus aureus and Their Toxicity in Drosophila melanogaster"

_molecules, 2020, doi:10.3390/molecules25092103_

Round 1
Reviewer 1 Report
The manuscript was well written and the data supports the conclusions. However, there are several areas the authors should consider to dramatically improve the impact and readability of the manuscript.
Major points
- The authors should include a figure of the structures of the molecules used in the study (Thymol, carvacrol, tetracycline, etbr).
- The authors should add to the discussion about the mechanism of action and resistance to EtBr and Tetracycline. This will help the readers evaluate the mechanism of the thymol and carvacrol.
- The authors should expand the discussion to include regarding other combinatorial approaches for antimicrobial action. Numerous groups have looked at combinations of antibiotics, and should be discussed
- It is unclear what "air" is referring to. The toxicity assay refers to 15ug/ml of air (page 7, line 222; also in figures), which must be defined.
Minor Points
- Several figures have Thymol referred to as "Timol".
Author Response
The manuscript was well written and the data supports the conclusions. However, there are several areas the authors should consider to dramatically improve the impact and readability of the manuscript.
Major points
- The authors should include a figure of the structures of the molecules used in the study (Thymol, carvacrol, tetracycline, etbr).
Answer: Thank you for your comment. The structures were included and the methodology was described.
- The authors should add to the discussion about the mechanism of action and resistance to EtBr and Tetracycline. This will help the readers evaluate the mechanism of the thymol and carvacrol.
Answer: Thank you for your comment. The discussion section was improved accordingly.
- The authors should expand the discussion to include regarding other combinatorial approaches for antimicrobial action. Numerous groups have looked at combinations of antibiotics, and should be discussed
Answer: The discussion section was changed accordingly.
- It is unclear what "air" is referring to. The toxicity assay refers to 15ug/ml of air (page 7, line 222; also in figures), which must be defined.
Answer: The term was explained in the text.
Minor Points
- Several figures have Thymol referred to as "Timol".
Answer: Corrected
Reviewer 2 Report
The manuscript „Evaluation of the antibacterial activity and efflux pump reversal of Thymol and Carvacrol against Staphylococcus aureus and their toxicity in Drosophila melanogaster by Zildene et al., describes here the antibacterial activity and efflux pump reversal mechanisms of the isomers thymol and carvacrol against the S. aureus IS-58 bacterial strain and further evaluated their toxicity in a D. melanogaster model.
Minimum Inhibitory Concentration (MIC) – How many times the test were performed, any statistical data
Line 47: change to- Among these pumps, the TetK pump belonging to the Major Facilitator Superfamily (MFS) is present in the S. aureus IS-58 strain. TetK powers its transport activity with energy derived from proton gradients and is responsible for resistance to the tetracycline class of antibiotics.
Line 109: change to- Furthermore, the association with EtBr demonstrated presence of an efflux pump mechanism sensitive to both standard inhibitors, which shows that the antibacterial activity exhibited by thymol and carvacrol against S. aureus are not due to an EPI.
Line 112: change to- To evaluate the role of thymol and carvacrol, two main components of O. vulgare essential oil, as antibiotic resistence modulators, agar dilution asssay was performed in presence and absence of sub-inhibitory concentrations (1/4 MIC) of the compounds using S. aureus strain IS-58, which contains efflux mechanism for tetracyclin. It was observed that both compounds modulated tetracyclin activity with MIC reduction from 64ug/mL to 32ug/mL in both cases.
Line 172: change to- The toxicity mechanisms and locomotor capacity impairments produced by many compounds and/or essential oil may be associated with a decrease in AChE acitivity which, considering its importance against neurotoxicity, functions as defense in stressful situations.
Line 176:change to- … heat shock proteins such as hsp70 which support the functional structure…
Line 188: What ratio DMSO:Water was used?
Author Response
The manuscript „Evaluation of the antibacterial activity and efflux pump reversal of Thymol and Carvacrol against Staphylococcus aureus and their toxicity in Drosophila melanogaster by Zildene et al., describes here the antibacterial activity and efflux pump reversal mechanisms of the isomers thymol and carvacrol against the S. aureus IS-58 bacterial strain and further evaluated their toxicity in a D. melanogaster model.
Minimum Inhibitory Concentration (MIC) – How many times the test were performed, any statistical data.
Answer: This information was added in the revised version of the manuscript.
Line 47: change to- Among these pumps, the TetK pump belonging to the Major Facilitator Superfamily (MFS) is present in the S. aureus IS-58 strain. TetK powers its transport activity with energy derived from proton gradients and is responsible for resistance to the tetracycline class of antibiotics.
Answer: Done
Line 109: change to- Furthermore, the association with EtBr demonstrated presence of an efflux pump mechanism sensitive to both standard inhibitors, which shows that the antibacterial activity exhibited by thymol and carvacrol against S. aureus are not due to an EPI.
Answer: Done
Line 112: change to- To evaluate the role of thymol and carvacrol, two main components of O. vulgare essential oil, as antibiotic resistence modulators, agar dilution assay was performed in presence and absence of sub-inhibitory concentrations (1/4 MIC) of the compounds using S. aureus strain IS-58, which contains efflux mechanism for tetracyclin. It was observed that both compounds modulated tetracyclin activity with MIC reduction from 64ug/mL to 32 ug/mL in both cases.
Answer: Done
Line 172: change to- The toxicity mechanisms and locomotor capacity impairments produced by many compounds and/or essential oil may be associated with a decrease in AChE acitivity which, considering its importance against neurotoxicity, functions as defense in stressful situations.
Answer: Done
Line 176:change to- … heat shock proteins such as hsp70 which support the functional structure…
Answer: Done
Line 188: What ratio DMSO: Water was used?
Answer: Done
Reviewer 3 Report
In this study, both thymol and carvacrol presented antibacterialactivities with and toxicity against D. melanogaster.
I don't have much to say about the experiments that show me well conducted
and supported by a correct statistical analysis. However, the results
don't seem very new to me. You also understand how the results are
discussed. The authors present their data as confirmations of other works.
Authors should make greater efforts to discuss their data by making
it clear why their work brings news
Author Response
In this study, both thymol and carvacrol presented antibacterial activities with and toxicity against D. melanogaster.
I don't have much to say about the experiments that show me well conducted and supported by a correct statistical analysis. However, the results don't seem very new to me. You also understand how the results are discussed. The authors present their data as confirmations of other works. Authors should make greater efforts to discuss their data by making it clear why their work brings news.
Answer: According to the reviewer comment, proper information was added in the discussion section.